# High Proportions of Radiation-Resistant Strains in Culturable Bacteria from the Taklimakan Desert

**DOI:** 10.3390/biology11040501

**Published:** 2022-03-24

**Authors:** Yang Liu, Tuo Chen, Juan Li, Minghui Wu, Guangxiu Liu, Wei Zhang, Binglin Zhang, Songlin Zhang, Gaosen Zhang

**Affiliations:** 1State Key Laboratory of Cryospheric Sciences, Northwest Institute of Eco-Environment and Resources, Chinese Academy of Sciences, Lanzhou 730000, China; liuyang21@nieer.ac.cn (Y.L.); chentuo@lzb.ac.cn (T.C.); wumh2017@lzb.ac.cn (M.W.); zhangbl@lzb.ac.cn (B.Z.); 2Key Laboratory of Extreme Environmental Microbial Resources and Engineering, Lanzhou 730000, China; liugx@lzb.ac.cn (G.L.); ziaoshen@163.com (W.Z.); 3University of Chinese Academy of Sciences, No. 19A Yuquan Road, Beijing 100049, China; 4College of Agriculture and Forestry Sciences, Qinghai University, Xining 810016, China; ljshamo@lzb.ac.cn; 5Key Laboratory of Desert and Desertification, Northwest Institute of Eco-Environment and Resources, Chinese Academy of Sciences, Lanzhou 730000, China; 6College of Geography and Environment Science, Northwest Normal University, Lanzhou 730070, China; zhangsonglin65@nwnu.edu.cn

**Keywords:** Taklimakan desert, culturable bacteria, UV-C, γ-rays, radiation-resistant extremophiles

## Abstract

**Simple Summary:**

Radiation-resistant extremophiles have frequently been found in the Taklimakan Desert, which is known for its harsh conditions. However, there is no systemic study investigating the diversity and proportion of radiation-resistant strains among culturable bacteria. The results of this study revealed the distribution of culturable bacteria in the Taklimakan Desert and indicated high proportions of radiation-resistant strains in the culturable bacteria. The study helps to better understand the ecological origin of radio-resistance and to quantitatively describe the desert as a common habitat for radiation-resistant extremophiles.

**Abstract:**

The Taklimakan Desert located in China is the second-largest shifting sand desert in the world and is known for its harsh conditions. Types of γ-rays or UV radiation-resistant bacterial strains have been isolated from this desert. However, there is no information regarding the proportions of the radiation-resistant strains in the total culturable microbes. We isolated 352 bacterial strains from nine sites across the Taklimakan Desert from north to south. They belong to Actinobacteria, Firmicutes, Proteobacteria, and Bacteroidetes. The phylum Actinobacteria was the most predominant in abundance and Firmicutes had the highest species richness. Bacteroidetes had the lowest abundance and was found in four sites only, while the other three phyla were found in every site but with different distribution profiles. After irradiating with 1000 J/m^2^ and 6000 J/m^2^ UV-C, the strains with survival rates higher than 10% occupied 72.3% and 36.9% of all culturable bacteria, respectively. The members from Proteobacteria had the highest proportions, with survival rates higher than 10%. After radiation with 10 kGy γ-rays, *Kocuria* sp. TKL1057 and *Planococcus* sp. TKL1152 showed higher radiation-resistant capabilities than *Deinococcus radiodurans* R1. Besides obtaining several radiation-resistant extremophiles, this study measured the proportions of the radiation-resistant strains in the total culturable microbes for the first time. This study may help to better understand the origin of radioresistance, especially by quantitatively comparing proportions of radiation-resistant extremophiles from different environments in the future.

## 1. Introduction

Microorganisms with the ability to survive high doses of radiation are known as radioresistant or radiation-resistant extremophiles. Radiation-resistant extremophiles are scattered in the three domains of life and have been isolated from diverse environments [1,2,3,4], which indicates that the radiation-resistant extremophiles do not have a common ancestor and do not have a specific ecological habitat. Since high doses of radiation were not encountered throughout most of life’s evolutionary history, radioresistance is thought to be a byproduct of resistance to other common stressors such as desiccation, oxidation, or heavy metals [5,6,7,8].

Within the varied environments, deserts are the most common places to find radiation-resistant extremophiles. From the Sonoran Desert, extensive diversity of ionizing-radiation-resistant bacteria has been found and isolates of the genera *Deinococcus*, *Geodermatophilus*, and *Hymenobacter* have been recovered after exposure to doses of 17 to 30 kGy γ-rays [9]. Paulino-Lima and his colleagues also isolated some extremely high-UV-C-radiation-resistant microorganisms from the Sonoran Desert and the Atacama Desert [10,11]. From the Sahara Desert, the type strains of radiation-resistant species *Deinococcus deserti*, *Geodermatophilus tzadiensis*, *Geodermatophilus sabuli*, and *Geodermatophilus pulveris* were isolated and identified [12,13,14]. The desert cyanobacteria, especially the genus *Chroococidiopsis*, are highly resistant to ionizing radiation [15]. Two *Deinococcus thermus* strains that were resistant to a dose of >600 J/m^2^ UV-C and >15 kGy of gamma radiation were isolated from the Lut desert of Iran, the hottest place on Earth [16].

In the Taklimakan Desert, radiation-resistant bacteria have also been isolated. Many of them were identified as novel taxa, including the type strains of *Hymenobacter xinjiangensis*, *Deinococcus taklimakanensis*, *Deinococcus xinjiangensis*, *Radiobacillus deserti*, *Streptomyces taklimakanensis*, and *Desertibacter roseus* [17,18,19,20,21,22]. There were also radiation-resistant yeasts isolated from the Taklimakan Desert, such as the strains XJ5-1 and 13-2 of *Aureobasidium melanogenum*, which produce melanin or macromolecular pullulan to help them survive in stressful environments [23,24]. Yu and his colleagues [4] investigated the diversity of ionizing-radiation-resistant bacteria in the Taklimakan Desert by isolating bacteria from samples that were irradiated by 3KGy γ-rays, and isolated 53 strains of γ-rays radiation-resistant bacteria belonging to the genera *Agrococcus*, *Arthrobacter*, *Cellulomonas*, *Deinococcus*, *Knoellia*, *Kocuria*, *Lysobacter*, *Microvirga*, *Nocardioides*, *Paracoccus*, *Planomicrobium*, *Pontibacter*, and *Rufibacter*.

However, all the previous studies focused on isolating the radiation-resistant extremophiles only, and the information about the proportions of the radiation-resistant strains in the total culturable microbes was not measured. This is not helpful in quantitatively describing the best ecological habitats for the radiation-resistant extremophiles to better understand the origin of radioresistance. Some researchers have hypothesized that “radiation resistance is a byproduct of adaptation to desiccation” to explain that the desert is common place to find radiation-resistant extremophiles. However, this has not been proven. Beblo-Vranesevic and colleagues found that there was no correlation between desiccation and radiation tolerance in microorganisms from diverse extreme environments tested under anoxic conditions [25]. More likely, the ROS-scavenging system is the interlinkage that enables the microbes to resist many different stresses [26,27,28], including desiccation and radiation. If this is true, the photochemical production of reactive oxygen species in desert soils [29] can explain the reason for the high likelihood of finding radiation-resistant strains in the desert.

To better understand the ecological niche for radiation-resistant extremophiles, we propose to determine the proportions of the radiation-resistant strains in total culturable bacteria frequencies to find the radiation-resistant strains or abundance of the known radiation-resistant-associated genes in metagenomes from different environments. In this study, when we investigated the diversity and distribution of culturable microorganisms from the Taklimakan Desert, we exposed the isolates to high doses of UV-C for screening the radiation-resistant microorganisms and evaluating the proportions of the radiation-resistant strains in the total culturable microbes to provide comparable data between different environments.

## 2. Materials and Methods

### 2.1. Site Description and Sample Collection

The Taklimakan Desert covers an area of 337,600 km^2^ and 85% of the surface is occupied by active dunes, which makes it the second-largest mobile desert in the world and the largest desert in China [30]. The Taklimakan Desert is hyper-arid, with <25 mm year^−1^ long-term mean annual precipitation [31] and 1000 mm of potential evapotranspiration. It has a very windy climate, with high solar radiation (http://data.cma.cn, accessed on 20 June 2020). The temperature fluctuates widely from −20 °C to 70 °C.

With intervals of 40 km and a distance of 500 m from the road, nine sampling sites were selected along with the desert road inside the Taklimakan Desert (Figure 1). At each site, triple samples with a depth of 0~5 cm were collected individually from different slope aspects of the dunes in September 2018. The samples were sealed in Labplas sterile bags (EFR-5590E, TWIRL’EM^®®^, Mississauga, ON, Canada) and stored at −20 °C until analyzed.

### 2.2. Measurement of Soil Physico-Chemical Properties

The water content (WC) was calculated by the D-value (difference of moisture content before and after drying), as weighting 20 g fresh soil samples in aluminum boxes before and after being dried at 105 ℃ [32]. The soil pH was measured using a pH meter (PT-10, Sartorius, Göttingen, Germany) with the ratio of fresh soil:water = 1:2.5 (*w/v*) [33]. The electrical conductivity (EC) was measured by the conductivity meter (DDSJ-308A, Leici, Shanghai, China) following the ratio of fresh soil:water = 1:5 (*w*/*v*). The contents of total nitrogen (TN), total carbon (TC), and total organic carbon (TOC) were measured by an element analyzer (Elementar Vario-EL, Langenselbold, Hessen Germany) [34,35,36]. For each sample, three parallel measurements of each physico-chemical property were needed.

### 2.3. Microbial Enumeration

Five grams of soil was mixed with 10 mL sterile saline solution (0.9% NaCl in distilled water) and sterile glass beads in a sterile flask. After shaking in the orbital shaker with 200 rpm at 30 ℃, the suspension was serially diluted. Then 100 μL of each diluted suspension was plated on the agar media of LB (1.0% tryptone, 0.5% yeast extract, 1.0% NaCl, 1.5% agar), R2A (0.05% yeast extract, 0.05% tryptone, 0.05% casamino acid, 0.05% glucose, 0.05% soluble starch, 0.03% K_2_HPO_4_, 0.005% MgSO_4_·7H_2_O, 0.03% CH_3_COCOONa, 1.5% agar, pH 7.2), and TSB (1.5% tryptone, 0.5% soya peptone, 0.5% NaCl, pH 7.2) for the isolation and enumeration of culturable bacteria. For each mixture of different dilutions, at least three parallel agar media plates were utilized to meet the minimum numbers for statistical analyses. After the colonies were formed, strains were purified on the LB agar media with the streak plate method, cultivated with 10 mL liquid LB medium, and then stored at −80 ℃ freezer in a solution of medium supplemented with glycerol at a final concentration of 20 % (*v*/*v*).

### 2.4. 16S rRNA Gene Sequencing and Phylogenetic Analyses

The total genomic DNA of the bacterial isolates was extracted using the Bacterial DNA Extraction Kit (Omega Bio-Tek, Norcross, GA, USA) according to its manufacturer’s instructions. The 16S rRNA gene fragment was amplified using the primers 27F and 1492R [37]. The PCR reaction cycling condition was as follows: initial denaturation at 94 °C for 5 min, followed by 30 cycles of denaturation at 95 °C for 30 s, annealing at 55 °C for 30 s, elongation at 72 °C for 45 s, and final elongation at 72 ℃ for 10 min. PCR products were purified and sent to Qinke (Qinke Company, Xi’an, China) for sequencing by the Applied Biosystems 3730XL (ABI 3730XL) sequencer. With the sequencing results, the strains were identified using the EzTaxon-e server [38]. The neighbor-joining trees were recovered in MEGA 6.0 [39]. The sequences were aligned with ClustalW [40]. Kimura’s two-parameters model was utilized to calculate evolutionary distances [41]. Confidence in the topologies of the resultant tree was calculated by the bootstrap test with 1000 resamplings [42].

### 2.5. Screening Radiation-Resistant Strains

Before the final testing doses of 1000 J/m^2^ and 6000 J/m^2^ UV-C were determined to use for irradiation screening, some isolated strains in this study were irradiated by doses ranging between 100–600 J/m^2^, as reported before, to screen radiation-resistant extremophiles in the pre-experiments [10,11]. Many of the strains had survival rates higher than 90%. Therefore, we chose 10 times the reported dose [16] to screen the UVC-resistant extremophiles. Strains in axenic culturing were grown in liquid LB medium to the later log phase with 200 rpm at 30 ℃. The turbidity of cell suspensions was measured with a spectrophotometer at the wavelength of 600 nm. The cell suspensions were adjusted to OD600 = 1 with liquid LB medium and then divided into 3 aliquots. One was reserved for control without receiving radiation, and it was serially diluted with 10^3^~10^4^ folds and plated on LB and R_2_A agar media. The other two aliquots were exposed to the UV-C irradiation with doses of 1000 J/m^2^ and 6000 J/m^2^, respectively, and then diluted with 10~10^4^ folds and plated on LB and R_2_A agar media. As a comparison with the reported data, *Escherichia coli* DSM 30,083 and *Deinococcus radiodurans* R1, which were purchased from the German collection of microorganisms and cell cultures (DSMZ), were used as the negative control and the positive control, respectively. The plates were incubated at 28 ℃ for 3~7 days and the colonies were counted. The survival rates were calculated. Within the 352 tested strains, the 7 strains with the best survival rates were exposed to γ-rays radiation from a 57 Co/Rh source at a dose of 10 KGy, and *Deinococcus radiodurans* R1 was used as the control.

### 2.6. Data Analysis

The map of the sampling sites was drawn by ArcGIS version 10.5 [43]. The one-way analysis of variance (ANOVA) was performed using SPSS 24.0. The software R 3.6.1 was used to perform Spearman analysis to find the correlation between the environmental factors and the abundance of each bacterial species. The network of the correlations was visualized with the software Cytoscape (version 3.7.0) [44]. The other diagrams were drawn by Origin Pro 2017 [45]. The relationship between soil properties (pH, EC, BD, SWC, and C/N) and the proportions matrix of the populations with different survival rates (with intervals at 20%) was determined by redundancy analysis (RDA) using CANOCO (ver. 4.5, Plant Research International, Wageningen, Netherlands) and the significance of the effect of each variable was defined using Monte Carlo permutation tests (permutations = 999). The resulting significance level was tested by the F- and *p*-values [46].

## 3. Results

### 3.1. The Physico-Chemical Properties of Sand Soil from the Taklimakan Desert

The altitudes of the sampling sites ranged from 868 m to 1155 m above sea level. The results from Table 1 show that the samples contained very little water, with values ranging between 0.118–0.276%. Within them, only two sites on the north side had WC values higher than 0.2%. The conductivities varied between 248–733 S/m. The pH values of the samples decreased from 8.92 to 7.92 from north to south in the Taklimakan Desert. The TN had a similar trend, decreasing from 1.352% to 0.668 % (Table 1). The TC varied between 1.135–2.21% with lower values in the middle of the desert, and the values of TOC were between 0.351–0.733% (Table 1).

### 3.2. Abundance and Diversity of Culturable Bacteria

The total numbers of culturable bacteria ranged between 3.3 × 10^3^–5.1 × 10^5^ CFU/g and varied significantly among the sites. At the phylum level, the abundance of the culturable Actinobacteria decreased from north, 2.47 × 10^5^ CFU/g, to south, 1.71 × 10^5^ CFU/g, namely from site T1 to T9 (Figure 2). Firmicutes and Proteobacteria had similar decreasing trends, with their abundance varying between 0.2 × 10^5^–3.5 × 10^5^ CFU/g and 5.9 × 10^4^–2.24 × 10^5^ CFU/g, respectively. Bacteroidetes were found in four sites only, namely T2–T5, with abundances of less than 3 × 10^4^ CFU/g.

Based on the morphology of the colonies, 2216 isolates were picked from all the agar plates with different media. After purification with restreaking the isolates on agar R_2_A plates, identical isolates were merged and finally 352 strains were selected for further analyses. The 16S rRNA gene sequencing results were deposited in GenBank with the accession numbers MW827797-MW828147 and MW835718. BLAST of the isolates’ 16S rRNA gene sequences with the Eztaxon database (EzBioCloud.net. Searching the closet type strains of valid species on 29 March 2019) showed that the 352 strains belonged to 4 phyla and 43 genera with different distributions in the nine sampling sites (Figure 3 and Appendix A). The phyla Firmicutes and Actinobacteria had higher species richness, while the phylum Bacteroidetes had only two strains that belonged to *Pontibacter* and *Sphingobacterium* (Figure 3). The phylum Actinobacteria had 20 genera in total and the dominant species were the members of the genera *Arthorbacter*, *Citricoccus*, *Kocuria*, *Microbacterium*, *Streptomyces*, *Nocardioides*, and *Saccharothrix* (Appendix A). At the genus level, *Bacillus* had the highest number of species members, 129 strains, and occupied 84% of the strains of the phylum Firmicutes. *Streptomyces* and *Pseudomonas* were the second and third largest member occupant genera and had 70 and 26 strains, respectively. The genera *Arthrobacter*, *Nocardioides*, *Kocuria*, *Saccharothrix*, *Gracilibacillus*, *Altereythrobacter*, and *Pseudochrobactrum* had more than five strains (Figure 3).

### 3.3. Distribution of Culturable Bacteria and Correlations with Environmental Factors

Phylogenetic analyses with 16S rRNA gene sequences grouped the 352 strains into 96 phylotypes, and the 16S rRNA gene sequence data of type strains of each phylotype were used to reconstruct the phylogenetic trees (Appendix A). The distribution profiles of the 96 phylotypes were also combined into Appendix A. The phylum *Actinobacteria* had the most diverse phylotypes, in total 44 phylotypes. Within them, *Streptomyces* had 11 phylotypes (Appendix A). The phylotypes in the phyla Firmicutes, Actinobacteria, and Proteobacteria were found in every site, but the phylotypes belonging to phylum Bacteroidetes were found in four sites only, namely the sites T2 to T5. At the genus level, the *Bacillus* and the *Streptomyces* were ubiquitously distributed in all the sites (Appendix A). The genus *Pontibacter* was detected in site T2 only and *Sphingobacterium* was found in sites T3–T5 (Appendix A).

Phylotype TKL-G1, affiliated with *Bacillus halotolerants*, was found in every site (Appendix A). Some phylotypes were found in northern sites (T1–T5), such as TKL239, TKL101, TKL-G13, TKL-G14, and TKL-G37, affiliated with *Bacillus solisilvae*, *Bacillus zhanjiangensis*, *Gracilibacillus dipsosauri*, *Gracilibacillus ureilyticus,* and *Chelativorans multitrophicus*, respectively (Appendix A). Some phylotypes were found in southern sites only, for instance, TKL1080 and TKL930, affiliated with *Rothia mucilaginosa* and *Shigella flexneri*, respectively (Appendix A).

Spearman’s correlation results showed that the environmental factors were significantly correlated with the abundance of some phylotypes from the phyla Actinobacteria, Firmicutes, and Proteobacteria (Figure 4). Within all the analyzed strains, 95 strains (43, 37, and 15 strains in Actinobacteria, Firmicutes, and Proteobacteria, respectively) were significantly correlated with one or several environmental factors. The 43 Actinobacteria strains had 34 significant positive correlations and 16 significant negative correlations. TOC had 18 significant positive correlations and 11 significant negative correlations, WC had 23 significant positive correlations and 11 significant negative correlations, and pH had 26 significant positive correlations and 8 significant negative correlations. Within the phylum *Actinobacteria*, 27% of the strains showed a significant correlation with TOC, TC, and WC contents. Within the phylum *Firmicutes,* most strains were significantly correlated with EC, pH, TOC, and TN contents, and the strains with the larger abundance had a higher significant correlation with EC and pH contents (Figure 4).

### 3.4. Survival Rates after Exposure to UV-C and γ-rays Radiation

In our study, the survival rates of the positive control, *Dinococcus radiodurans* R1, were 42.1% after 1000 J/m^2^ UV-C radiation (low-dose UV-C radiation, abbreviated as LR thereafter) and 31.6% after 6000 J/m^2^ UV-C radiation (high-dose UV-C radiation, abbreviated as HR thereafter), while those of the negative control, *Escherichia coli*, were 10% under LR and 0.42% under HR, respectively. After radiating all isolates with 1000 J/m^2^ and 6000 J/m^2^ UV-C, the strains with survival rates higher than 10% occupied 72.3% and 36.9% of all culturable bacteria, respectively (Figure 5). Within the tested 352 strains, 253 strains had survival rates higher than 10% under LR. In other words, the D_10_-values (10% survival dose) of these 253 strains were higher than 1000 J/m^2^ UV-C radiation. Under HR, 134 strains had survival rates higher than 10%. Namely, their D_10_-values were higher than 6000 J/m^2^ UV-C radiation. At the phylum level, the survival rates of all members from the phylum Bacteroidetes were lower than those of the negative control, while the proportions of the strains with survival rates higher than 10% were Proteobacteria > Firmicutes > Actinobacteria under HR. The ranks changed with different ranges of survival rates (Figure 5A). Under LR, Firmicutes had the highest proportions of the strain, with survival rates between 20–50% and higher than 80%, while Proteobacteria was the highest, with survival rates between 50–80%. Actinobacteria had the highest proportions, with survival rates between 10–20% (Figure 5A). Redundancy analysis (RDA) results indicated that only TC contents had a significant impact on the proportion matrix, with different survival rates under 6000 J/m^2^ UV-C radiation (Figure 6B). T4 and T6 had higher proportions of the population with survival rates between 60–80% and over 80%, respectively (Figure 6). Another interesting finding is that almost all the strains with a better radiation-resistance ability than *Dinococcus radiodurans* R1 were isolated from the central area. For example, TKL1057, the strongest UV-C resistant strains in this study, were isolated from the sites T4, T5, T6, and T7 (Appendix A).

To show the details of the UV-C-resistant capabilities of the culturable bacteria in the Taklimakan Desert, the strains with higher survival rates than *Escherichia coli* are shown in Figure 7. In the phylum Actinobacteria, 15 strains had higher survival rates under LR than *Dinococcus radiodurans* R1. The TKL1057, closely related to the species *Kocuria indica*, had the highest survival rates, namely 92.3% under LR and 87.2% under HR (Figure 7A). In the phylum Firmicutes, 10 strains had higher survival rates under LR than the positive control. Within them, the strain TKL1152 (*Planococcus citreus*) had the strongest ability for UV-C radiation-resistance. Moreover, the survival rate of TKL1152 was far higher than that of other strains in Firmicutes (Figure 7B). In the phylum Proteobacteria, only five strains had higher survival rates under LR than the positive control. The strongest four strains in Proteobacteria, TKL865 (*Paracoccus hibisci*), TKL35 (*Altererythrobacter soli*), TKL855 (*Pseudochrobactrum saccharolyticum*), and TKL606 (*Pseudomonas xanthomarina*), had similar survival rates under both LR and HR (Figure 7C).

The survival rates of all the strains under HR decreased when compared with their survival rates under LR, but they did not decrease by equal amounts or proportions, leading to great changes between the survival rates under HR and LR. For example, in the phylum Actinobacteria, TKL860 *Citricoccus zhacaiensis* and TKL897 *Streptomyces griseoviridis* had the third and second highest survival rates under HR, respectively, but their survival rates under LR were much lower (Figure 7A). However, the survival rate rankings for the most resistant member in each phylum, namely TKL1057 *Kocuria indica*, TKL1152 *Planococcus citreus*, and TKL865 *Paracoccus hibisci*, did not change (Figure 7).

*Dinococcus radiodurans* R1 had 13% survival cells after 10KGy γ-rays radiation; these results are the same as those reported previously [47]. With the same dose of γ-rays radiation, the strains TKL1057 and TKL1152 had higher survival rates than *Dinococcus radiodurans* R1 (Figure 8). The strains TKL606 and TKL865 had survival rates of around 12%, similar to *Dinococcus radiodurans* R1s. TKL855, TKL897, and TKL860 had survival rates of 11.62%, 9.3%, and 6.2%, respectively (Figure 8).

In addition, the survival rates assay showed that six isolates with strong radiation-resistance ability against both UV-C and γ-rays were pigment-generated with colors such as red, yellow, etc. (the colors of the pigmented isolates are provided in the Appendix A). Comparing the proportions of 23 pigment-generating strains in the 352 strains, the colored strains had a better radiation-resistance ability than non-colored strains.

## 4. Discussion

### 4.1. Diversity of Distribution of Culturable Bacteria in the Taklimakan Desert

To the best of our knowledge, our study is the first to systemically reveal the diversity and distribution of culturable bacteria in the central region of the Taklimakan Desert. Previous studies with pyrosequencing indicated that surface sand samples close to the Taklimakan Desert and the sandstorms originating there were dominated by four phyla—Proteobacteria, Bacteroidetes, Actinobacteria, and Firmicutes [48,49]. In our study, there were the same four phyla but the abundance ranks were very different. After pre-treatment with 3KGy γ-rays, Yu and his colleague isolated bacteria belonging to five phyla, the above four phyla, and Deinococcus-Thermus [4]. However, they used different media, modified TSB and PYFG, and isolated much fewer strains than we did. Some of the literature in Chinese reported that the dominant bacteria were from Firmicutes and Proteobacteria in and around the Taklimakan Desert [50,51,52]. In our study, the phylum Actinobacteria had the highest species richness and was the most widely distributed across the Taklimakan Desert.

In similar environments of shifting sand desert, the dunes of the subtropical coastal forest were dominated by Acidobacteria and Proteobacteria [53], and the tropical coastal dunes were dominated by Actinobacteria, Proteobacteria, Chloroflexi, and Firmicutes [54]. The desert dune in Qinghai–Tibet had very similar diversity, with the dominant phyla being Firmicutes, Actinobacteria, Proteobacteria, and Bacteroidetes [55]. In most deserts, high-throughput sequencing results indicated that the phyla Bacteroidetes, Firmicutes, and Proteobacteria were the dominant bacteria, but, in terms of culturable bacteria, the members of the phylum Actinobacteria were the predominant species [56,57,58,59,60]. For instance, in our study, there were more Gram-positive bacteria, members of the phyla Firmicutes and Actinobacteria, than Gram-negative bacteria in cultured bacteria in the Taklimakan Desert. However, in a similar ecological environment, in the Thar Desert, there were more Gram-negative bacteria, members of the phyla Proteobacteria and Bacteroidetes, than Gram-positive bacteria in cultured bacteria such as members of the phyla Firmicutes and Actinobacteria [61].

The relative abundance of culturable bacteria in the Taklimakan Desert were ranged between 10^3^–10^5^ CFU/g, which generally were 2–4 orders of magnitude less than those in agricultural soils [62,63] and two orders of magnitude less than those in the Sahara and Gibson Deserts [64], but similar to those in dunes [65,66], which might be explained by the nutrient levels. In our study, the abundances of culturable bacteria were decreased from the north to the south, which is consistent with the total nitrogen contents. Moreover, the CFUs of the phylum Actinobacteria presented a regular decrease from site T1 to T9 (Figure 2), where the pH values had the same changes, namely decreasing from north to south in the Taklimakan Desert. Abundance of Actinobacteria is correlated with pH value [67]. The pH value of sand soil from each site that affected the distribution of Actinobacteria was also observed in other deserts [67,68]. Besides the pH value, the presence of the phylum *Actinobacteria* was also positively correlated with carbon and nitrogen contents in many studies by high-throughput sequencing analysis [69,70,71].

### 4.2. The Proportions of Radiation-Resistant Strains in the Culturable Bacteria

One of the main purposes of this study was to attempt to find the most probable ecological habitat of radiation-resistant bacteria via determining the proportions of the radiation-resistant strains in the total culturable bacteria and comparing this index between different environments. In the Taklimakan Desert, 72.3% of the culturable bacteria had D10-values higher than 1000 J/m^2^ UV-C radiation, and 36.9% culturable bacteria had D10-values higher than 6000 J/m^2^ UV-C radiation (Figure 5). The proportions of the radiation-resistant strains in the total culturable bacteria appear to be quite high, yet we could not compare these proportions with previous studies because there were no reports as regards the proportions of the radiation-resistant strains in the culturable bacteria from the other environments. Previous reports regarding radiation-resistant bacteria strains were mainly to isolate the strain with the resistant capability only. We believe this measurement could be a good indicator to reveal the most probable ecological habitat of radiation-resistant bacteria. In the future, if we have more data about this proportion in varied environments, we may be able to establish the ecological habitat characteristics of the environments after comparison.

Within this study, we compared the profiles of cultivable bacteria from different sites. The results indicate that the hinterland sites had lower proportions of the population with lower survival rates but higher proportions of the population with higher survival rates—for example, the site T4 versus T9. The RDA results indicated that the sites T4 and T6 had higher proportions of the population with survival rates between 60–80% and over 80%, respectively. In addition, the RDA results indicate that TC was a significant factor that had a negative correlation with a higher proportion of survival rates, which might be explained by “more nutrients, less stress” (Figure 6). Nevertheless, due to the low heterogeneity of environmental factors within the Taklimakan Desert, the comparison within the sites is not ideal. We expect to find more interesting results when there are more investigations of the proportions from other environments.

At the phylum level, the relative proportions of the strains with survival rates higher than 10% were Proteobacteria > Firmicutes > Actinobacteria under HR, which contradicts the common-sense view of drought-resistant capability. In previous reports, members from Firmicutes or Actinobacteria were able to tolerate aridity better than those from Proteobacteria [72,73,74]. Under harsh environmental conditions, members of the phyla Actinobacteria and Firmicutes can withstand aridity because of their sporulation or protective dormancy [30,75,76]. In addition, it has been reported that UV-induced DNA damage in Gram-positive bacteria is lower than in Gram-negative bacteria because of a shielding action by the cell wall [77]. This topic deserves to be studied further.

Here we should mention that the doses of UV-C radiation tested in our study were approximately 10-fold higher than those in some previous reports [4,11]. The reason might be due to the different radiation methods. Many previous studies irradiated UV-C after the plating with the cells. Hence, the dose of UV-C radiation per cell might have been different. We suggest adjusting the cell concentrations to a similar level, such as the ~1 OD used in this study, to standardize the radiation energy received for each cell. This standardized measurement would provide advantages in comparing different studies. For example, the γ-rays radiation results of the positive and negative controls were similar to those from other studies because the unit of Gy is a unit of measurement for absorbed radiation per biomass [78,79].

### 4.3. The Traits of the Strongly Radiation-Resistant Bacteria in the Taklimakan Desert

Within the strains from the Taklimakan Desert, seven strains had a resistance capability that was similar to or even higher than that of *Dinococcus radiodurans* R1. Within them, two Gram-positive bacterial strains showed higher survival rates than *Dinococcus radiodurans* under UV-C and γ-rays irradiation by directly radiating the culturable bacteria using UV-C and γ-rays. These are the TKL1057 strains, belonging to the genus *Kocuria*, and T1152 strains, belonging to the genus *Planococcus*. In previous reports, in the genus *Planococcus*, the strains *Planococcus citreus* DSM 20549, *Planococcus maritimus* JCM 11543, and *Planococcus dechangensis* DSM 25,871 were strongly radiation-resistant bacteria [38,80,81]. In the genus *Kocuria*, several strains from the species *Kocuria rhizophila* and *Kocuria rosea* were reported as radiation-resistant [82,83,84].

An interesting finding is that almost all the strains with a radiation-resistance ability greater than or similar to that of *Dinococcus radiodurans* were isolated in the T4, T5, T6, and T7 sites (Appendix A), all of which were located in the central parts of the Taklimakan Desert. They therefore had longer annual sunshine durations and exposure to more solar irradiation than those from other sites (http://data.cma.cn, accessed on 20 June 2020). This suggests that, rather than the extreme physico-chemical properties of the soil, the key factor in the eco-environment for finding more strongly radiation-resistant bacteria may be the sunshine duration [85].

In addition, strains with a strong radiation-resistance ability against both UV-C and γ-rays were pigment-generated. Comparing the proportion of 6.5% pigment-generated strains in the 352 strains, the strongest seven strains had 85.7% pigment-generated strains (Figure 5 and Appendix A). For instance, strains TKL1152 and TKL865 could produce red pigments. Moreover, strains TKL1057 and TKL606 could produce gold pigments. Another strong ability of radiation-resistant strains TKL855, TKL860, and TKL897 was that they could produce pink, orange, and purple pigments, respectively. The phenomenon was also observed by the other researchers with different experiments. Some previous studies have reported that, after irradiation, those bacteria that were found with certain kinds of pigments were those that had the greatest radiation-resistant ability [81,86,87], which explains how the pigments of bacteria may absorb irradiation to protect the cells from ionization damage. For example, the deinoxanthin of carotenoids, which could eliminate ROS after UV-C radiation from the secondary metabolites of *Dinococcus radiodurans* R1, could protect cells from the damage of UV-C radiation [88,89].

## 5. Conclusions

The Taklimakan Desert has a very low biomass but a quite high diversity of culturable bacterial. The 352 bacterial strains belonged to 4 phyla, with 43 genera having different distributions in the nine sites across the Taklimakan Desert from north to south. The phylum Actinobacteria was the most abundant, while Firmicutes had the highest species richness. Bacteroidetes had the lowest abundance and was found in the north only. The abundance and distribution of the culturable bacteria were affected by the physiochemical traits of the soil. In the culturable bacteria, there were very high proportions of the strains with radiation-resistant capabilities: 72.3% of the culturable bacteria had D_10_ values of UV-C higher than 1000 J/m^2^ and 36.9% of the strains’ D_10_ values were higher than 6000 J/m^2^. The members from Proteobacteria generally had the highest survival rates after irradiation with UV-C. Within the isolates from the Taklimakan Desert, eight strains had resistance capabilities that were similar to or even greater than those of *Dinococcus radiodurans* R1. After radiation with 10kGy γ-rays, *Kocuria* sp. TKL1057 and *Planococcus* sp. TKL1152 showed higher radiation-resistant capability than *Dinococcus radiodurans* R1. Besides providing new radiation-resistant extremophile strains, this study provided the proportions of the radiation-resistant strains in the total culturable microbes for the first time, which is helpful to better understand the origin of radio-resistance and to quantitatively describe the desert as a common habitat for radiation-resistant extremophiles.

## Figures and Tables

**Figure 1 biology-11-00501-f001:**
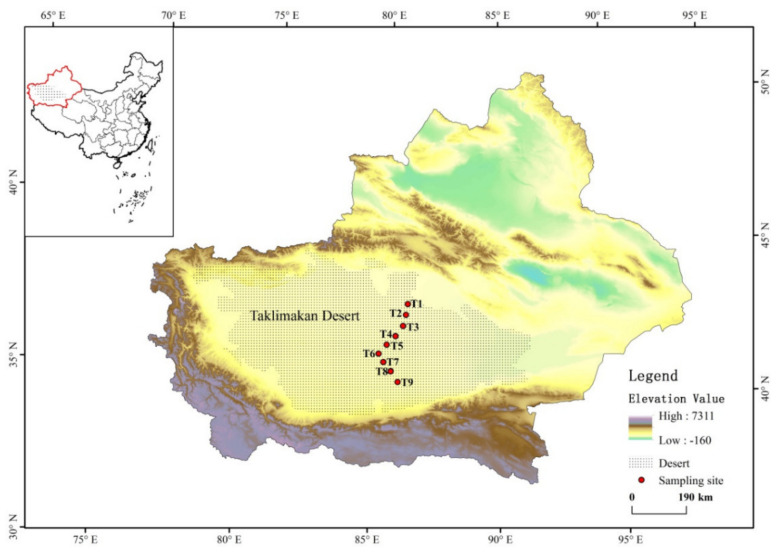
The sampling sites of the Taklimakan Desert, northwestern China.

**Figure 2 biology-11-00501-f002:**
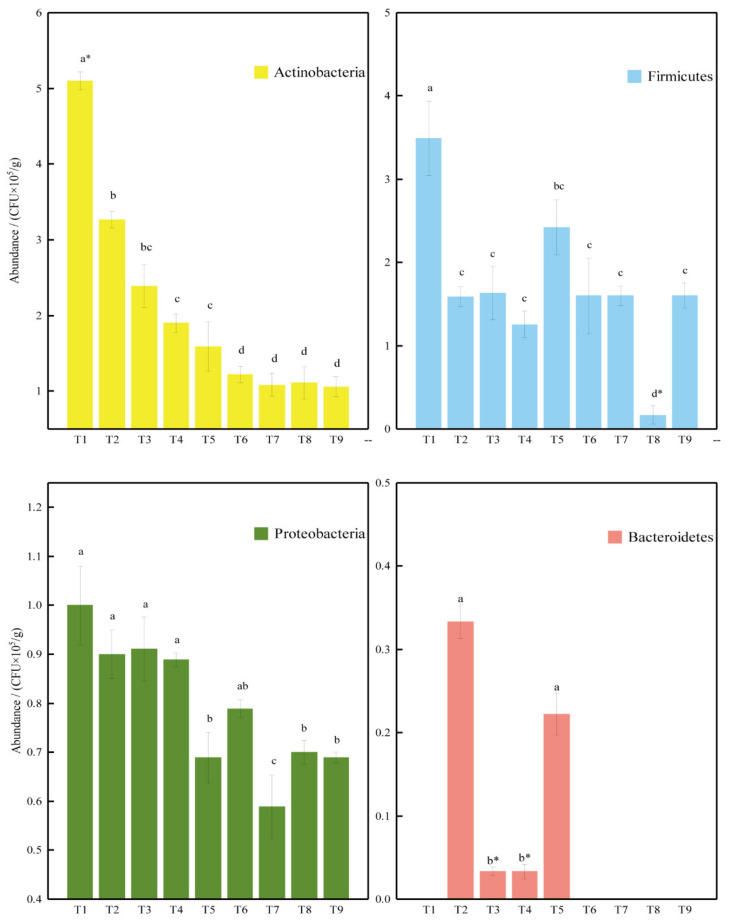
The numbers of culturable bacteria (mean ± SD) of the nine sampling sites. For each graphic, different lower-case letters among site treatments indicate significant differences (*p* < 0.05, Tukey’s HSD tests), while the symbol * refers to *p* < 0.01.

**Figure 3 biology-11-00501-f003:**
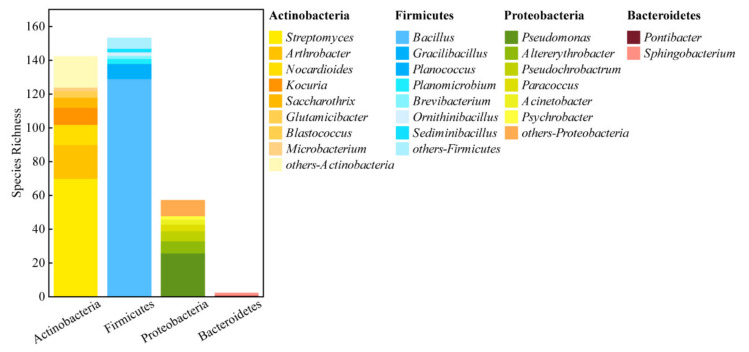
Species richness of the 4 phyla at the genus level. The different columns of each bar with different colors represent different genera, and the height of each column with different colors represents the species richness.

**Figure 4 biology-11-00501-f004:**
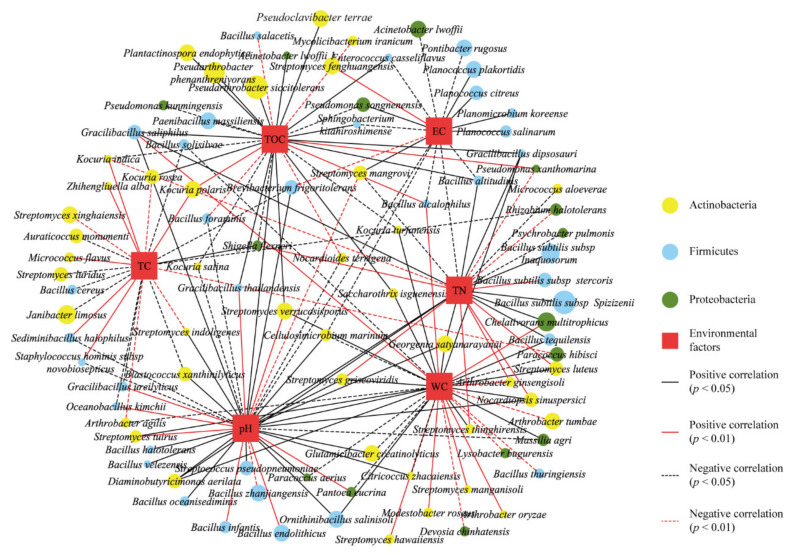
Co-occurrence networks of culturable bacterial species associated with environmental factors. A node refers to a species and the size of each node indicates the abundance of the species. Each edge means significant correlations between the nodes. Within the edges, the solid line and the dashed lines represent positive and negative correlations, respectively, while the black and red lines refer to Spearman’s correlation at the levels of *p* < 0.05 and *p* < 0.01, respectively.

**Figure 5 biology-11-00501-f005:**
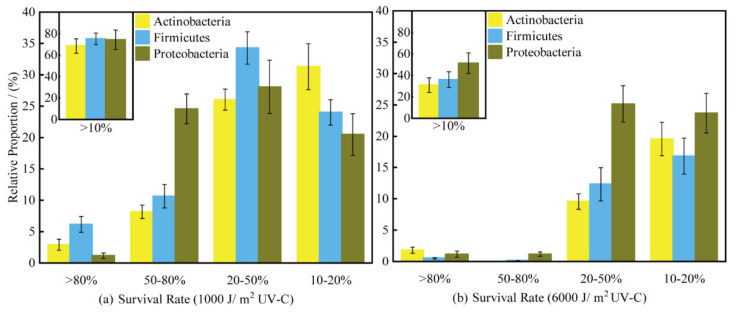
The proportions of the radiation-resistant strains in total culturable bacteria under the different doses of radiation in each phylum: (**a**) after irradiation with a low radiation of 1000 J/m^2^ (LR); (**b**) after a high radiation of 6000 J/m^2^ (HR) of UV-C radiation.

**Figure 6 biology-11-00501-f006:**
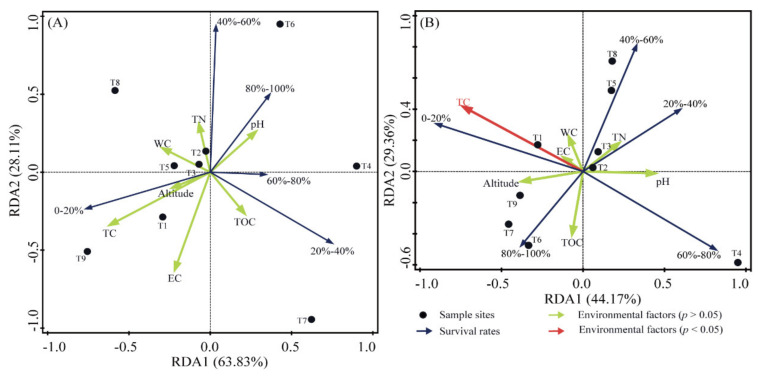
The correlation analyses among the sample sites, environmental factors, and survival rates matrix with a low radiation of 1000 J/m^2^ (**A**) and a high radiation of 6000 J/m^2^ (**B**). WC, water content; EC, electrical conductivity; TN, total nitrogen; TC, total carbon; TOC, total organic carbon.

**Figure 7 biology-11-00501-f007:**
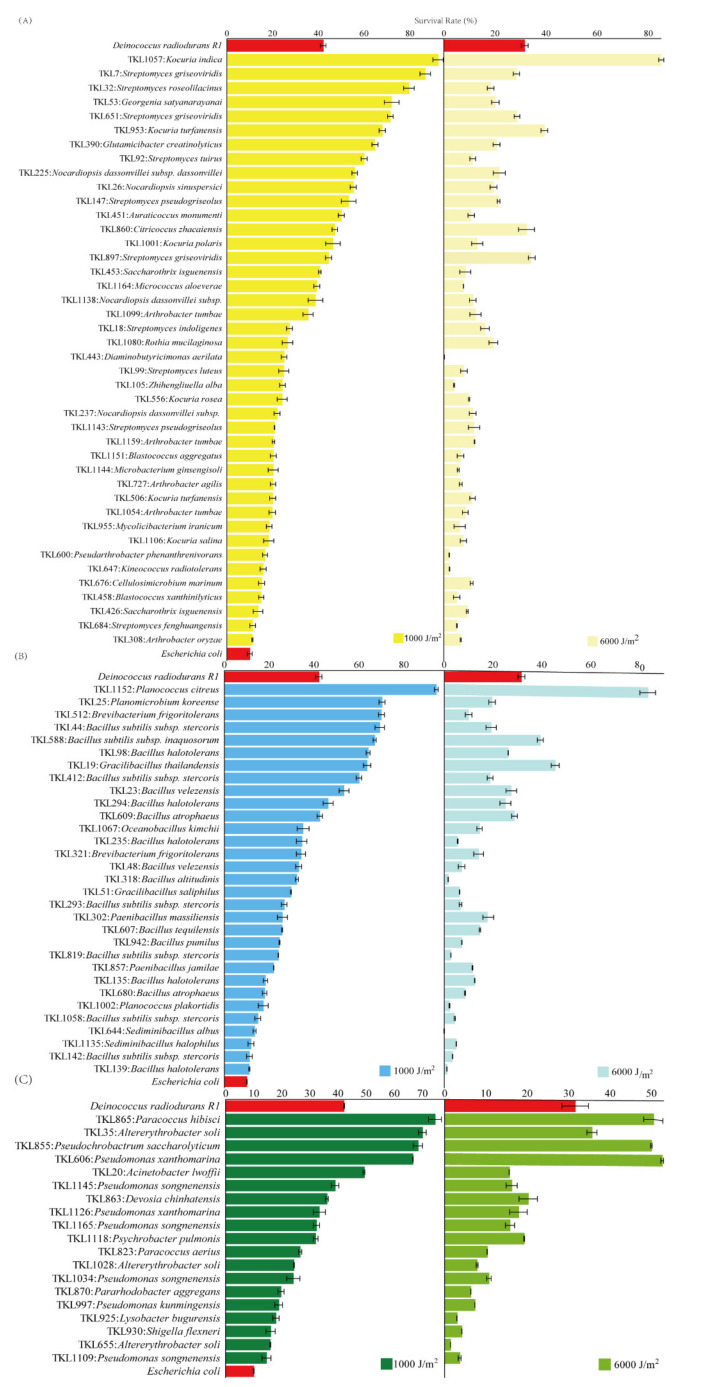
Viability of 44 isolates, 32 isolates, and 21 isolates affiliated with the phyla *Actinobacteria* (**A**), *Firmicutes* (**B**), and *Proteobacteria* (**C**), respectively, after irradiation with a low radiation of 1000 J/m^2^ (LR) and a high radiation of 6000 J/m^2^ (HR) of UV-C radiation. The survival values of *Dinococcus radiodurans* and *Escherichia coli* were used as controls and are shown for comparison. The survival rates of all members from the phylum Bacteroidetes were less than the negative control and were not displayed in this figure.

**Figure 8 biology-11-00501-f008:**
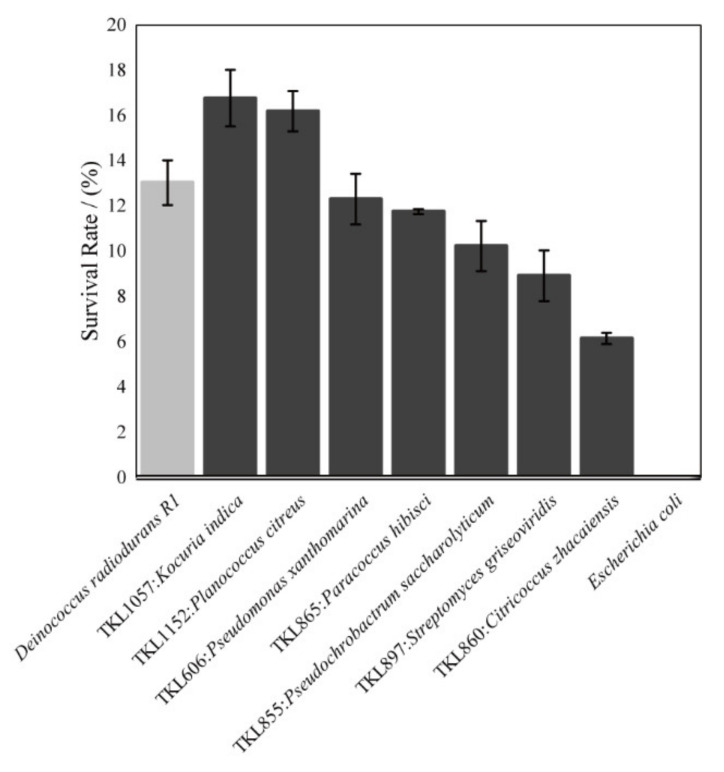
Viability of the 8 most UV-C resistant bacterial strains after irradiation with a 10 KGy dose of γ-rays irradiation. The *Dinococcus radioduran* R1 was used as a control.

**Table 1 biology-11-00501-t001:** The physicochemical properties (mean ± SD) of the desert samples.

Samples	Altitude (m)	WC (%)	pH	EC (S/m)	TN (%)	TC (%)	TOC (%)
T1	868 ± 3.2	0.218 ± 0.010	8.58 ± 0.11	310 ± 0.12	1.352 ± 0.11	2.321 ± 0.12	0.732 ± 0.016
T2	876 ± 3.2	0.276 ± 0.020	8.92 ± 0.18	386 ± 2.23	1.352 ± 0.11	2.122 ± 0.25	0.712 ± 0.008
T3	901 ± 3.2	0.118 ± 0.008	8.62 ± 0.24	297 ± 0.58	1.221 ± 0.05	1.655 ± 0.31	0.733 ± 0.012
T4	938 ± 3.2	0.145 ± 0.020	8.67 ± 0.21	458 ± 0.45	1.022 ± 0.15	1.135 ± 0.15	0.659 ± 0.037
T5	982 ± 3.2	0.153 ± 0.020	8.82 ± 0.13	633 ± 0.34	0.998 ± 0.08	1.685 ± 0.08	0.652 ± 0.034
T6	1045 ± 3.2	0.162 ± 0.012	8.43 ± 0.18	248 ± 1.22	0.998 ± 0.09	1.674 ± 0.09	0.632 ± 0.034
T7	1068 ± 3.2	0.134 ± 0.025	8.01 ± 0.22	671 ± 1.34	0.668 ± 0.02	2.122 ± 0.08	0.598 ± 0.019
T8	1080 ± 3.2	0.185 ± 0.008	7.92 ± 0.31	496 ± 1.38	0.889 ± 0.02	2.135 ± 0.05	0.351 ± 0.034
T9	1155 ± 3.2	0.165 ± 0.010	7.92 ± 0.18	733 ± 0.38	0.768 ± 0.08	2.134 ± 0.01	0.653 ± 0.016

Note: WC, water content; EC, electrical conductivity; TN, total nitrogen; TC, total carbon; TOC, total organic carbon.

## Data Availability

The datasets generated for this study can be found in GenBank under the accession numbers MW835366-MW835717 and MW835718.

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
