# Peer review of "High Proportions of Radiation-Resistant Strains in Culturable Bacteria from the Taklimakan Desert"

_biology, 2022, doi:10.3390/biology11040501_

Round 1
Reviewer 1 Report
The work presented by Liu et al. studies the bacterial diversity and radiation resistance (UVC and γ-rays) of the culturable fraction from the Taklimakan Desert. To achieve this goal, several phylogenetic analyses and experiments have been carried out. The study overall provides some interesting points and valuable data. However, the results section needs to be more accurate and requires further analyses.
Broad comments
* The phylogenetic trees presented in this work are not accurate and they should be redone.
* This work lacks some key analyses to describe the species richness of the different samples (i.e. Chao). These analyses should be performed .
* The resistance experiments should include the Bacteroidetes.
Specific comments
Line 23: Where is this desert? Please, provide its location here.
Line 46: Please, replace "3" by three.
Lines 76-80: This sentence is too long and difficult to follow. Please, split it up into two or more sentences.
Line 91: To determine the proportions or frequencies of what? Please, clarify this point.
Line 126: Please, replace here and throughout the text "rmp" by "rpm" (rpm is an abbreviation for “revolutions per minute”).
Line 137: 16S rRNA
Line 147-148: "Confidence on the topologies of the resultant tree...": Bootstrap values indicate the confidence of a given topology.
Line 151: “turbidity”
Line 144: Please, provide more information on how the sequencing was carried out (i.e. sequencer, sequencing kit).
Line 178: Please, replace “were increased” by "ranged".
Lines 200-203: To deposit sequences in GenBank does not indicate any microbial diversity result. Please, describe how this analysis was performed to assign the phyla and genera to the colonies observed.
Line 204-212: The term "species richness" is misleading as this analysis was not performed. Also it merely takes into account the number of genera present for each phyla, not the species.
Line 212: "more than 5 strains": This is not shown in figure 3. Please, clarify this point.
Line 215: Which were the treatments? Please, clarify this point.
Figure 3: What do the numbers in the Y axis represent? What represent the bars in these figures?
Line 222: The phylogenetic trees presented in this work are not accurate since members of Bacillus were present in the tree assigned to Actinobacteria. Also members belonging to the genus Streptomyces were not shown in any tree.
The phylogenetic analysis needs to be redone. Given the importance of these trees in the results and discussion sections they should be included within the main text.
Line 239: This figure is misleading as too many information is included therein. Please, redo it by displaying the data for each phylotype in a different figure.
Lines 264, 283-295, Figure 5 and Figure 6: In this resistance experiments, what about Bacteroidetes? Where are the member of this important phyla? They were also observed in the phylogenetic analysis. To better complete the resistance profile of the culturable fraction they should be included.
Lines 269-275: The data referred in these sentences are shown in Figure 5, not 6. Please, clarify this point.
Figure 5: Error bars should be included.
Lines 281-282: Please, indicate in this figure legend the difference between A and B.
Line 274, 386, 388: Where are these RDA results indicated? What is the meaning of RDA?
Figure 6: What represent the bars in these figures?
Lines 302-305: This information is not present in Figure 5.
Line 306: Replace “champions” by “the more resistant”.
Line 329: Reference 48 does not provide the referred information. Please, remove it.
Figure 8, line 380: This analysis should go into the results section.
Line 414: “7 strains”
Reviewer 2 Report
Dear authors, in general, the manuscript you wrote made a good impression. However, it seems to me that some points require clarification.
- Do the authors assume that the identified microbiota is characteristic of a certain climatic season? If yes, then this should be emphasized in the discussion and abstract.
- Solar radiation plays a big role in the topic chosen by the authors and the conclusions of the study made by them. In this regard, I consider it important to give a quantitative description of the insolation of sampling sites in the article. This could help researchers compare UV-resistant bacteria reports from different locations.
-
Why were UV doses of 1000 J/m2 and 6000 155 J/m2 selected for testing bacteria for UV resistance? What works did the authors rely on when choosing these doses? Add to the text of the article.
-
Methodological note. Three rich culture media for the isolation of bacteria from soil samples have much in common in their composition. Bacteria growing on them have similar ecological characteristics. The number and diversity of isolated species could be increased by using more different media, for example by adding a medium with the mineral form of nitrogen.
-
Obviously, a mistake was made in the additional materials. The same dendrogram was copied onto the first and second pages.
Reviewer 3 Report
The manuscript describes the isolation of bacteria from soil collected in the Taklimakan Desert. The resistance of selected isolated to UV is evaluated and the diversity of culturable bacteria is correlated to the physicochemical characteristics of the sampled sites. The manuscript is interesting however the poor quality of english and the colloquial style employed by the authors does not allow the manuscript to be accepted in its present version. Additionally the manuscript should be shortened and repetitions should be avoided between Results and Discussion. Here I present some selected examples of poor english and coloquial style, that should be improved in order to accept the manuscript:
- The “Simple summary” (lines 16-22) is deficient because it only gives general and colloquial information.
- The sentence “Many radiation-resistant bacterial strains were isolated from this desert” in line 24 should be rewritten in a different tense i.e. “have been isolated” or do the authors refer to what they obtained in the present work? The word many is too imprecise.
- Deinococucus instead of Dinococcus in all the text (line 35).
- The tenses of “indicated” (line 47) and “was” (lines 48 and 49) should be changed. Do not instead of don’t in line 48. This kind of English mistakes is found throughout the manuscript.
- This title is incorrectly written: “2.416. S rRNA Gene Sequencing and Phylogenetic Analyses”
- It is incorrect to write “axenic strains” as in line 150. A strain is axenic by definition, axenic culture would be correct.
- This title is incorrectly written: “3.1. The Physic-chemical Properties of the Taklimakan Desert”
- In line 446, Pretty high is too colloquial for a scientific manuscript.
Round 2
Reviewer 1 Report
The revised version of the work presented by Liu et al. has been improved according to previous comments, but not all.
Overall, I still have several concerns about this paper.
- The work still lacks some key analyses to describe the species richness of the different samples (i.e. Chao).
- The resistance experiments are still incomplete as they do not include the Bacteroidetes.
Author Response
Dear reviewer,
Thank you for your comments concerning our article “High proportions of radiation-resistant strains in the culturable bacteria from Taklimakan Desert” (Manuscript No.: biology-1616095). Please find our response followed your comments below.
Sincerely
Gaosen
19th-March-2022
Review1
Comments and Suggestions for Authors
The revised version of the work presented by Liu et al. has been improved according to previous comments, but not all.
Overall, I still have several concerns about this paper.
- The work still lacks some key analyses to describe the species richness of the different samples (i.e. Chao).
Answer: As we mentioned last time, we calculated the Chao and some other diversity indices, like the table below. However, we don’t think the results were very important and helpful to improve our manuscript because two reasons: (1) this study did not focus on the diversity indices; and (2) after all, very limited species are culturable and the “estimated species” here can be affected by the culture strategies a lot, hence the observed species richness (numbers of strains) is more accurate to describe the culturable bacteria.
I talked with my colleagues again and did not find a good solution to display the calculated diversity indices. Should we provide this table in the supplementary only?
|
Sites |
Chao 1 |
ACE |
Shannon (e) |
Simpson |
goods_coverage |
|
S1 |
65.3 |
90.9709 |
4.0398 |
0.8713 |
0.8244 |
|
S2 |
46.12 |
49.7406 |
4.9792 |
0.9531 |
0.9433 |
|
S3 |
52.71429 |
55.5040 |
5.1704 |
0.9677 |
0.8229 |
|
S4 |
47.5625 |
53.2471 |
5.0558 |
0.9638 |
0.8469 |
|
S5 |
31 |
33.3875 |
4.3496 |
0.9284 |
0.8800 |
|
S6 |
37 |
37.3467 |
4.3090 |
0.9375 |
0.8472 |
|
S7 |
38.09091 |
40.2995 |
4.8040 |
0.9581 |
0.8936 |
|
S8 |
23 |
24.5990 |
4.0080 |
0.9169 |
0.9296 |
|
S9 |
65.3 |
90.9709 |
4.0398 |
0.8713 |
0.8244 |
- The resistance experiments are still incomplete as they do not include the Bacteroidetes.
Answer: [As we mentioned last time, we did resistance experiments with members from the phylum Bacteroidetes. The survival rates of all strains from this phylum (specifically, there are only 2 strains, belonging to the genera Pontibacter and Sphingobacterium, respectively.) were less than the negative control, E. coli BL21, after the radiation of 1000 and 6000 J/ m2. Hence, we only include the results as statistical data in Fig. 5 and Fig. 6 (as mentioned in the original manuscript text, “Within the tested 352 strains,” (Line 284)) but did not include the results in the new Figure 7 (we should say sorry here, we said that “the results were not included in Fig. 5” in round 1). Now we added “the survival rates of all members from the phylum Bacteroidetes were less than the negative control” in Line 288-289.]

Round 3
Reviewer 1 Report
The revised version of the work presented by Liu et al. has been improved according to several previous comments, so I recommend its publication. Nevertheless, I have two more suggestions that may improve the paper:
- Given the importance of the phylogenetic trees in the results and discussion sections they should be included within the main text.
- Please, provide a comment in the Results section on the survival rates of strains from Bacteroidetes explaining why they were not present in the resistance experiments. The explanation given in the author response would be enough.